

# Genome-wide identification of the MADS-box gene family in *Avena sativa* and its role in photoperiod-insensitive oat

Jinsheng Nan[1], Jianghong An[1,2], Yan Yang[1], Guofen Zhao[1],
Xiaohong Yang[3], Huiyan Liu[1] and Bing Han[1]

[1] Inner Mongolia Agricultural University, Key Lab of Germplasm Innovation and Utlization of Triticeae Crop at Universities of Inner Mongolia Autonomous Region, Inner Mongolia Agricultural University, Hohhot, Inner Mongolia, China
[2] Inner Mongolia Academy of Agriculture and Animal Husbandry Science, Hohhot, Inner Mongolia, China
[3] Zhangjiakou Academy of Agricultural Sciences, Zhangjiakou, HeBei Province, China

Corresponding author
Bing Han, hb_nmg@163.com

## ABSTRACT

**Background:** Traditional spring-summer sown oat is a typical long-day crop that cannot head under short-day conditions. The creation of photoperiod-insensitive oats overcomes this limitation. MADS-box genes are a class of transcription factors involved in plant flowering signal transduction regulation. Previous transcriptome studies have shown that MADS-box genes may be related to the oat photoperiod.
**Methods:** Putative MADS-box genes were identified in the whole genome of oat. Bioinformatics methods were used to analyze their classification, conserved motifs, gene structure, evolution, chromosome localization, collinearity and *cis*-elements. Ten representative genes were further screened *via* qRT-PCR analysis under short days.
**Results:** In total, sixteen *AsMADS* genes were identified and grouped into nine subfamilies. The domains, conserved motifs and gene structures of all *AsMADS* genes were conserved. All members contained light-responsive elements. Using the photoperiod-insensitive oat MENGSIYAN4HAO (MSY4) and spring-summer sown oat HongQi2hao (HQ2) as materials, qRT-PCR analysis was used to analyze the *AsMADS* gene at different panicle differentiation stages under short-day conditions. Compared with HQ2, *AsMADS3*, *AsMADS8*, *AsMADS11*, *AsMADS13*, and *AsMADS16* were upregulated from the initial stage to the branch differentiation stage in MSY4, while *AsMADS12* was downregulated. qRT-PCR analysis was also performed on the whole panicle differentiation stages in MSY4 under short-day conditions, the result showed that the expression levels of *AsMADS9* and *AsMADS11* gradually decreased. Based on the subfamily to which these genes belong, the above results indicated that *AsMADS* genes, especially SVP, SQUA and Mα subfamily members, regulated panicle development in MSY4 by responding to short-days. This work provides a foundation for revealing the function of the *AsMADS* gene family in the oat photoperiod pathway.

## INTRODUCTION

Among the cereal crops, oat (*Avena sativa* L.) is cultivated worldwide as a food and forage crop. Traditional spring-summer sown oat is a long-day crop that cannot head without light in winter in the Northern Hemisphere (*Yang et al., 2018*). This traditional oat generally requires at least 14 h of sunlight per day to head (*An et al., 2018*). This feature of oats limits the expansion of the oat growing area. New photoperiod-insensitive germplasm was bred by "Interspecific Polymer Crossing Methods" and could complete the entire growth period and set seeds in natural short-day conditions in HaiNan, China (11 h light/ 13 h dark) (*Yang et al., 2014*). The creation of this new oat germplasm overcame the previous limitations of photoperiod-sensitive oat. However, the reason why photoperiod-insensitive oats can complete the growth period under short days has not been elucidated. Transcriptome studies have been conducted previously on photoperiod-insensitive materials and traditional oats, revealing that MADS genes may be related to the photoperiod insensitivity of oat (*An et al., 2020*).

As transcriptional regulatory factors, MADS-box proteins play a key role in controlling the signal transduction processes of plant flowering time, flower organ formation and meristem specificity (*Becker & Günter, 2003*). Hundreds of MADS-box genes have been found in many plants, such as Arabidopsis (*Parenicová et al., 2003*) maize (*Zhao et al., 2011*), rice (*Arora et al., 2007*), and wheat (*Susanne et al., 2020*). There are two types of MADS-box genes in plants, type I (SRF-like) and type II (MEF2-like, or MIKC type) (*Smaczniak et al., 2012*). Type I genes usually contain only one or two exons, with one highly conserved SRF-like MADS domain, and lack the K domain. Type II genes contain four conserved domain structures: the M domain (MADS-box), I domain (Intervening), K domain (Keratin-Like) and C-terminal domain (*Henschel et al., 2002*). Most of the MIKC-type MADS-box genes contain six introns and seven exons, called MIKC$^C$, while some contain seven introns and eight exons, called MIKC*. In Arabidopsis, the type I MADS-box genes encode three types of proteins, including Mα, Mβ and Mγ types (*Arora et al., 2007*). The MIKC$^C$ proteins can be subdivided into 13 distinct subfamilies (AG, AGL6, AGL12, AGL15, AGL17, DEF/GLO, BS, FLC, SEP/AGL2, SQUA, SVP, TM3/SOC1 and AP3) (*Theien, 2001*). The MIKC* type can be divided into the P type and S type. The P-type MIKC* proteins identified in Arabidopsis are AGL30, AGL65 and AGL94, and the S-type proteins are AGL66, AGL67 and AGL104 (*Kwantes, Liebsch & Verelst, 2012*).

The ABCDE model is a classical genetic regulatory mechanism of flower organ development in Arabidopsis (*Theien, 2001*). AP1 is an A-type gene whose functions include recognition genes that promote the formation of floral meristems and the development of sepals, petals and other floral organs (*Kempin, Savidge & Yanofsky, 1995*). The AP1/SQUA homologous gene VRN1 in wheat is a flowering activator and is regularly expressed in leaves under long- and short-day conditions. VRN1 is located upstream of FT, upregulates FT expression and promotes flowering under long-day conditions (*Shimada et al., 2009*). DEF/GLO is a B-class gene that controls the morphology of the floral organs such as stamens and petals (*Kai-Uwe et al., 2002*). In Arabidopsis, the formation of sepals, petals, stamens, carpels and ovules is related to the SEP gene, which forms a MADS protein

complex with A, B, C, and D proteins to work together in the morphogenesis of floral organs (*Favaro et al., 2003*). SVP-group MADS-box genes were first identified as flowering suppressor regulators in Arabidopsis. In rice, their roles in regulating meristem identity are well conserved, and their involvement in determining flowering time is not significant (*Lee, Jeong & An, 2008*).

To explore their role in the photoperiod insensitivity of oat, the MADS-box gene family was identified at the genome-wide level. In this study, 16 MADS members were identified in *Avena sativa*, and their phylogenetic relationship, gene structure, conserved motifs, chromosome localization and collinearity were analyzed, along with *cis*-elements in the promoter region of all identified MADS-box genes. Transcriptome data conducted in the initial stage of oat panicle differentiation in the traditional oat HongQi2hao (HQ2) and photoperiod-insensitive oat MSY4 under short days were used for differential expression analysis. In addition, the expression patterns of candidate genes at different panicle differentiation stages under short days were analyzed. Our results help to further reveal the biological function of MADS genes in photoperiod-insensitive oat and help to lay a foundation for revealing the mechanism of photoperiod insensitivity in oat.

## MATERIALS AND METHODS

### Identification of MADS genes in *Avena sativa* L

To identify the potential MADS-box genes in oat, the genome sequences were obtained from the GrainGenes database (https://wheat.pw.usda.gov/GG3/graingenes_downloads/oat-ot3098-pepsico). The MADS-box protein sequences of Arabidopsis (https://www.arabidopsis.org) and rice (http://rapdb.dna.affrc.go.jp/download/irgsp1.html) were downloaded. Hidden Markov Model-based searches (http://hmmer.janelia.org/), built from these known MADS-box protein sequences, were used to search the MADS-box proteins in oat and as queries to search against the oat protein databases with the BLASTP program with an e-value of $1 \times e^{-10}$ as the threshold. The software pfamscan and the PFAM A database were used to annotate the domains of the candidate *AsMADS* sequences, and the sequence containing the PF00319 domain was determined to be the final MADS-box sequence in oat. The information of sequence length, molecular weight, isoelectric points, and the instability index was obtained from the Expsay website (http://web.expasy.org/protparam/) (*Panu et al., 2012*). The subcellular localization was predicted using the Softberry website (http://www.softberry.com/).

### Sequence alignment and phylogenetic analysis

The MADS-box protein sequences of Arabidopsis (https://www.arabidopsis.org) and rice (http://rapdb.dna.affrc.go.jp/download/irgsp1.html) were downloaded and contained 108 and 75 sequences, respectively. Sequence alignment and evolution analysis were carried out on the MADS protein sequences identified in oat, Arabidopsis and rice. Mafft (https://mafft.cbrc.jp/alignment/software/) was used for protein multiple sequence alignment. DNAMAN software was used to align sequences and recheck data result. Both NCBI batch CDD searches and pfamscan were used to double check the conserved domains. TBtools software (https://bio.tools/tbtools) was used for visual display (*Chen et al., 2020*). MEGA

version 7 was used for neighbor-joining (NJ) tree construction based on the alignment of MADS proteins (*Kumar et al., 2018*). AsMADS genes were classified according to their phylogenetic relationships with the corresponding Arabidopsis (*Parenicová et al., 2003*) and rice (*Arora et al., 2007*) MADS genes.

## Gene structure, conserved motif analysis, *Cis*-element and protein structure analysis

Gene structure prediction was performed in GSDS (http://gsds.cbi.pku.edu.cn/). The conserved motifs were analyzed by MEME software (http://meme-suite.org/) (*Bailey et al., 2009*).The upstream regulatory regions (2 kb from the translation start site) of *AsMADS* genes were obtained from the whole genome sequence of *Avena sativa* by TBtools. The *cis*-elements on the promoter were predicted by the website http://plantregmap.gao-lab.org/. The positions of the binding sites in the physical map of the gene promoters are marked and displayed. Protein secondary structure was investigated by the Prabi website (https://npsa-prabi.ibcp.fr/cgi-bin/npsa_automat.pl?page=npsa_sopma.html), and the 3D structure was predicted on the SWISS-MODEL website (https://swissmodel.expasy.org/interactive).

## Chromosome localization and collinearity analysis

According to the position of *AsMADS* genes on the chromosome, MegaGene2Chrom (http://mg2c.iask.in/mg2c_v2.0/) was used to draw the chromosome physical location map. MCScanX software was used to perform family gene collinearity analysis based on chromosome length information and gene structural annotation information. An intraspecific collinearity map of oat and an interspecific collinearity map of oat, rice and Arabidopsis were constructed.

## Transcriptional profile analysis

For *AsMADS* gene expression analysis, RNA-seq data of *Avena sativa* L. materials with different photoperiod sensitivities under short-day conditions were obtained from previous laboratory data (*An et al., 2020*; *Becker & Günter, 2003*). Short-day treatment-related transcriptome analysis was conducted in the initial stage of panicle development in that study, in which 10 MADS genes were differentially expressed in the panicles and leaves of HQ2 and MSY4. Heatmaps were generated using these data. qPCR data results of 10 AsMADS genes was normalized by z-score. Heatmap is performed in Metware cloud platform tool (https://cloud.metware.cn/#/tools/tool-form?toolId=168). Each bar represents a gene. Red indicates that the gene is up-regulated, and blue indicates that the gene is down-regulated. GO function annotation of MADS genes (https://www.geneontology.org/).

## Plant materials and treatments

Similar to the materials used in the previous transcriptome study, MSY4 and HQ2 were used for specific expression analysis at different panicle developmental stages under short-days condition (12 h). The seeding and shading methods were described in previous studies (*Yang et al., 2018*; *An et al., 2018*, *2020*). The young panicles, including those at the initial stage, elongation stage, spikelet differentiation stage, floret differentiation stage,

pistil differentiation stage, and tetrad stage, were collected for expression analysis. Collected tissues were immediately frozen in liquid nitrogen and stored at −80 °C until RNA isolation was performed.

## Total RNA extraction and qRT–PCR expression analysis

The total RNA of oat panicles was extracted using a TransZol Up Plus RNA Kit (Transgene, Beijing, China) and then reverse transcribed into cDNA by a TransScript® One-Step gDNA Removal and cDNA Synthesis Kit. The integrity of the RNA was analyzed by agarose gel electrophoresis, and the concentration and purity of the RNA were detected by a NanoDrop 2000. The cDNA was stored at −20 °C for subsequent qRT-PCR. Specific primers for candidate genes in the MADS family were designed for qRT-PCR analysis using the Actin gene of oat as an internal reference gene (*Yang et al., 2013*). qRT-PCR was performed with SYBR-Green on a Jena Qtower 2.2 analyzer. According to the manufacturer's instructions, the 20 µL reaction contained 1 µL cDNA, 400 nM of each primer and 10 µL SYBR Green Mix. The amplification conditions were 95 °C for 1 min and 40 cycles of 95 °C for 10 s, 60 °C for 10 s and 72 °C for 10 s, with a melting curve over a temperature range of 60–95 °C. Each experiment used three biological replicates and three technical replicates. The relative expression levels of genes were calculated using the $2^{-\Delta\Delta Ct}$ method (*Schmittgen, 2008*). Gene-specific DNA primers for qPCR are listed in File S7.

## RESULTS

### Identification and characterization of the MADS-box gene family in *Avena sativa* L

Based on the oat genome (OT3098), a total of 16 MADS-box genes with the PF00319 domain were identified through the Hidden Markov Model, Blastp search and Pfam annotation, which were named *AsMADS1* to *AsMADS16*. Their basic properties were systematically evaluated, including gene ID and location, protein size, number of exons and introns, molecular weight, isoelectric points, instability index, GRAVY, and subcellular location (Table 1). The protein length of the *AsMADS* genes ranged from 142 to 278 amino acids, with 1 to 10 exons. The protein molecular weight ranged from 16.11 kDa (*AsMADS11*) to 31.61 kDa (*AsMADS2*), and the isoelectric points varied from 5.52 (*AsMADS3*) to 9.28 (*AsMADS16*). The hydrophobicity of the 16 proteins was negative. The instability index ranged from 44.82 to 65.59. The subcellular location prediction indicated that all members were located in the nucleus. The above results indicated that *AsMADS* genes were unstable hydrophobic nucleoproteins.

### Multiple sequence alignment and phylogenetic relationship analysis of MADS-box genes in *Avena sativa* L

A multiple sequence alignment analysis of the 16 AsMADS proteins was conducted using mafft. The N-terminus of the sequence was conserved (File S1). The 16 AsMADS protein sequences were analyzed for the presence of MADS-box and other domains using an NCBI batch CDD search. Our results showed that all the AsMADS proteins contained the M domain, and 13 out of 16 AsMADS proteins contained the K domain, except for

**Table 1 Information for the MADS gene family.**

| Gene name | Gene id | Location | Size (AA) | Exon | Intron | Molecular weight (KDa) | Isoelectric points | Instability index | GRAVY | Subcellular location |
|-----------|---------|----------|-----------|------|--------|------------------------|---------------------|-------------------|-------|----------------------|
| AsMADS1 | TRINITY_DN13687_c0_g2_i1.mrna1 | 1D:440731844-440737616 | 248 | 8 | 7 | 28.33 | 6.46 | 64.41 | −0.797 | Nuclear |
| AsMADS2 | TRINITY_DN231_c1_g1_i5.mrna2 | 3A:354464249-354472711 | 278 | 10 | 9 | 31.61 | 9.01 | 54.8 | −0.839 | Nuclear |
| AsMADS3 | TRINITY_DN32614_c0_g1_i3.mrna1 | 6C:205578143-205585989 | 221 | 7 | 6 | 24.64 | 5.52 | 49.5 | −0.634 | Nuclear |
| AsMADS4 | TRINITY_DN3261_c0_g2_i2.mrna1 | 7D:80735581-80737035 | 206 | 8 | 7 | 23.77 | 7.02 | 62.67 | −0.492 | Nuclear |
| AsMADS5 | TRINITY_DN3337_c0_g1_i3.mrna2 | 6D:221314556-221321673 | 263 | 9 | 8 | 29.83 | 8.9 | 45.02 | −0.776 | Nuclear |
| AsMADS6 | TRINITY_DN3733_c0_g1_i1.mrna1 | 2D:4056942-4066252 | 202 | 7 | 6 | 23.43 | 8.8 | 60.05 | −0.553 | Nuclear |
| AsMADS7 | TRINITY_DN383_c0_g1_i1.mrna1 | 6A:9765559-9768798 | 255 | 7 | 6 | 28.48 | 6.46 | 61.69 | −0.515 | Nuclear |
| AsMADS8 | TRINITY_DN383_c1_g1_i2.mrna1 | 1D:279668871-279671412 | 209 | 7 | 6 | 24.33 | 9.08 | 64.25 | −0.805 | Nuclear |
| AsMADS9 | TRINITY_DN39828_c0_g6_i1.mrna1 | 3A:87682568-87683651 | 182 | 1 | 0 | 20.14 | 6.76 | 54.46 | −0.416 | Nuclear |
| AsMADS10 | TRINITY_DN4009_c0_g1_i6.mrna1 | 1A:289587547-289592225 | 257 | 7 | 6 | 28.69 | 9.15 | 60.67 | −0.616 | Nuclear |
| AsMADS11 | TRINITY_DN42664_c0_g1_i9.mrna1 | 4D:391283530-391315386 | 142 | 5 | 4 | 16.11 | 9.09 | 65.59 | −0.723 | Nuclear |
| AsMADS12 | TRINITY_DN6792_c2_g1_i3.mrna1 | 4D:282480902-282492339 | 211 | 7 | 6 | 24.63 | 9.25 | 60.9 | −0.82 | Nuclear |
| AsMADS13 | TRINITY_DN6901_c0_g1_i1.mrna1 | 3D:340212580-340215197 | 196 | 7 | 6 | 22.32 | 8.67 | 44.82 | −0.59 | Nuclear |
| AsMADS14 | TRINITY_DN695_c0_g1_i7.mrna1 | 7A:351371284-351376726 | 233 | 7 | 6 | 27.03 | 8.96 | 45.71 | −0.845 | Nuclear |
| AsMADS15 | TRINITY_DN9392_c0_g1_i5.mrna2 | 2C:535820137-535822661 | 242 | 8 | 7 | 28.18 | 8.52 | 55.2 | −0.558 | Nuclear |
| AsMADS16 | TRINITY_DN9814_c0_g1_i1.mrna2 | 2C:39788836-39799997 | 278 | 8 | 7 | 31.51 | 9.28 | 59.65 | −0.856 | Nuclear |

AsMADS4, AsMADS9, and AsMADS11 (Fig. 1). Moreover, 14 out of the 16 members belonged to type II MADS, and the remaining two (AsMADS9 and AsMADS11) belonged to type I MADS (Fig. 1 and Files S2 and S3).

To evaluate the evolutionary relationships among the AsMADS proteins, a phylogenetic tree was constructed using a bootstrapped neighbor-joining (NJ) method with the amino acid sequences of 207 MADS genes (16 from oat, 108 from Arabidopsis, and 83 from rice). The phylogenetic analysis indicated that the 16 AsMADS members were grouped into nine subfamilies: SEP/AGL2, AGL6, SQUA, AGL12, AG, BS, SVP, Mα, and MIKC* (Fig. 2). No members were grouped into the TM3/SOC1, AGL17, AGL15, FLC or Mγ subfamilies. The results showed that the SVP subfamily contained 3 AsMADS members, the SEP/AGL2, SQUA, AG, BS and Mα subfamilies contained two members, and the remaining three subfamilies contained only one gene (Fig. 2 and File S3). In short, MADS

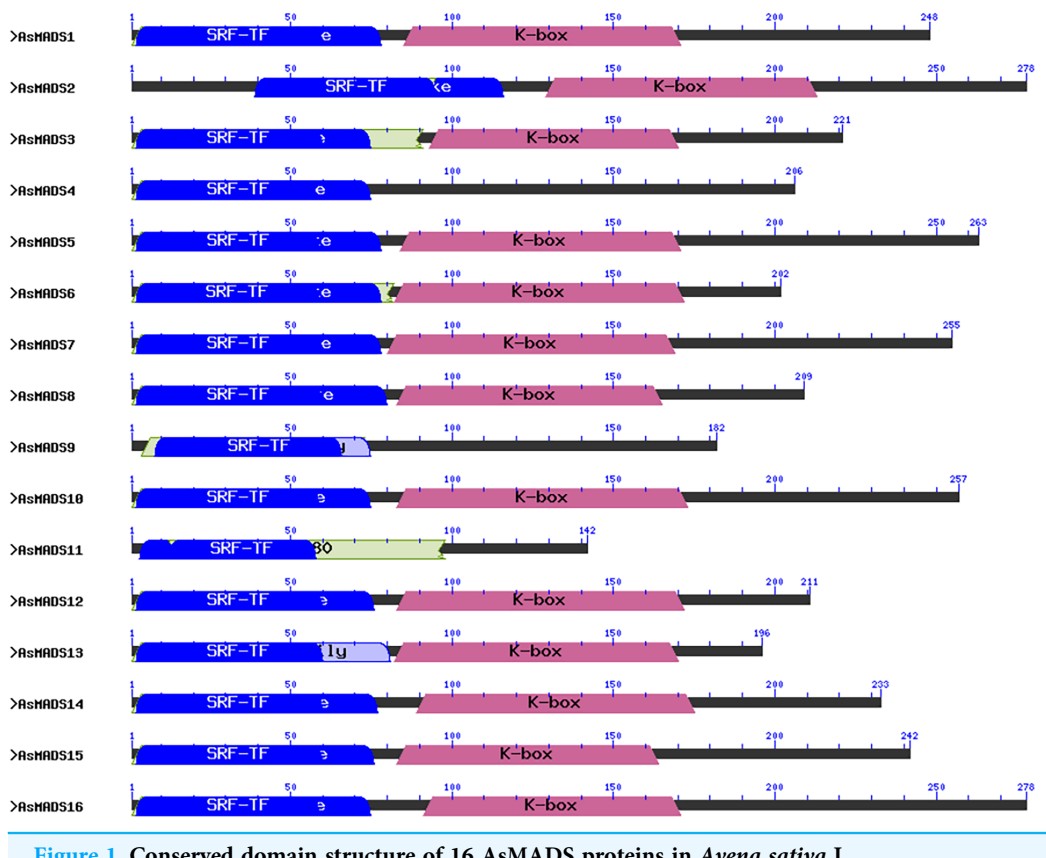

**Figure 1 Conserved domain structure of 16 AsMADS proteins in *Avena sativa* L.**

genes in oats consisted of type I genes (*AsMADS9* and *AsMADS11*), MIKC* (*AsMADS4*), and MIKC^c-type genes (13 other AsMADS members).

## Structural analysis and conserved motif composition of the *AsMADS* gene family

Gene structure prediction was performed in GSDS website (http://gsds.gao-lab.org/), which showed that 15 *AsMADS* members contained multiple introns and exons, with the number of exons ranging from five to 10 and the number of introns ranging from 4 to 9; *AsMADS9* only contained one exon (Table 1 and Fig. 3C). *AsMADS3, AsMADS8*, and *AsMADS13*, which were grouped in the SVP subfamily, had a similar intron/exon pattern, but *AsMADS9* and *AsMADS11* had radically different structures, even though they were grouped in the same subfamily.

The secondary structure of AsMADS proteins comprised an alpha helix, extended strand, beta turn, and random coil. The AsMADS proteins had a large proportion of alpha helix amino acids (>47%), followed by random coils (File S4). Except for AsMADS11, 15 of the 16 AsMADS proteins had similar 3D structures (Fig. 4), indicating similar functions. Compared with other genes, it is clear that the 3D predicted structure of AsMADS11 contains less α-helices and more β-sheets. This makes the 3D predicted structure of
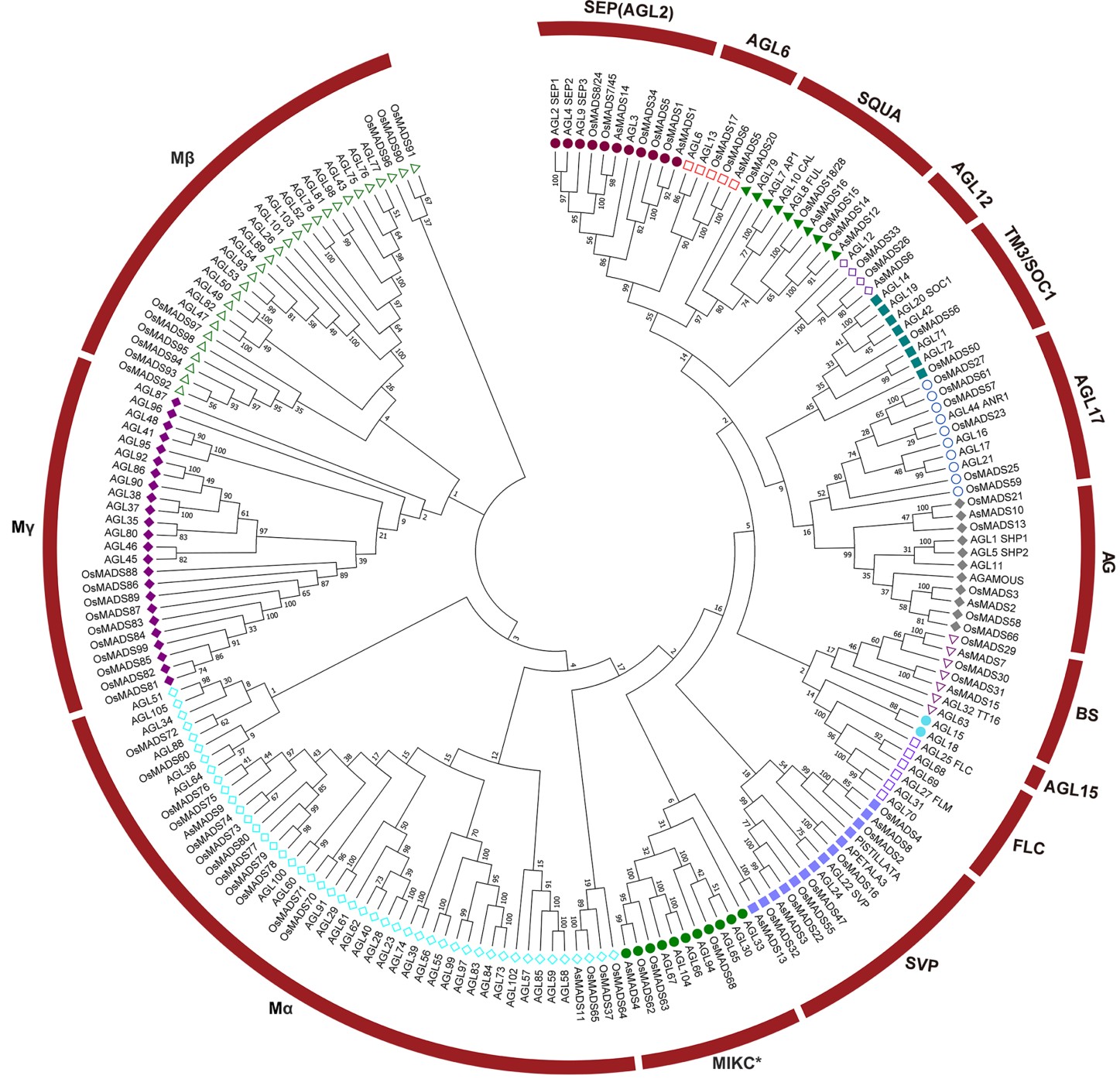

**Figure 2 A phylogenetic tree of the AsMADS gene family.** The MADS proteins of oat, Arabidopsis, and rice were clustered into 14 groups. Members of different colors belong to different groups.

AsMADS11 look simpler. However, the secondary structural pattern of AsMADS11 was similar to that of the other members (File S4).

The full-length protein sequences of 16 AsMADS were used to investigate the conserved motifs. In total, 15 different motifs were identified, and motifs 1, 2, 3 and 5 were widely

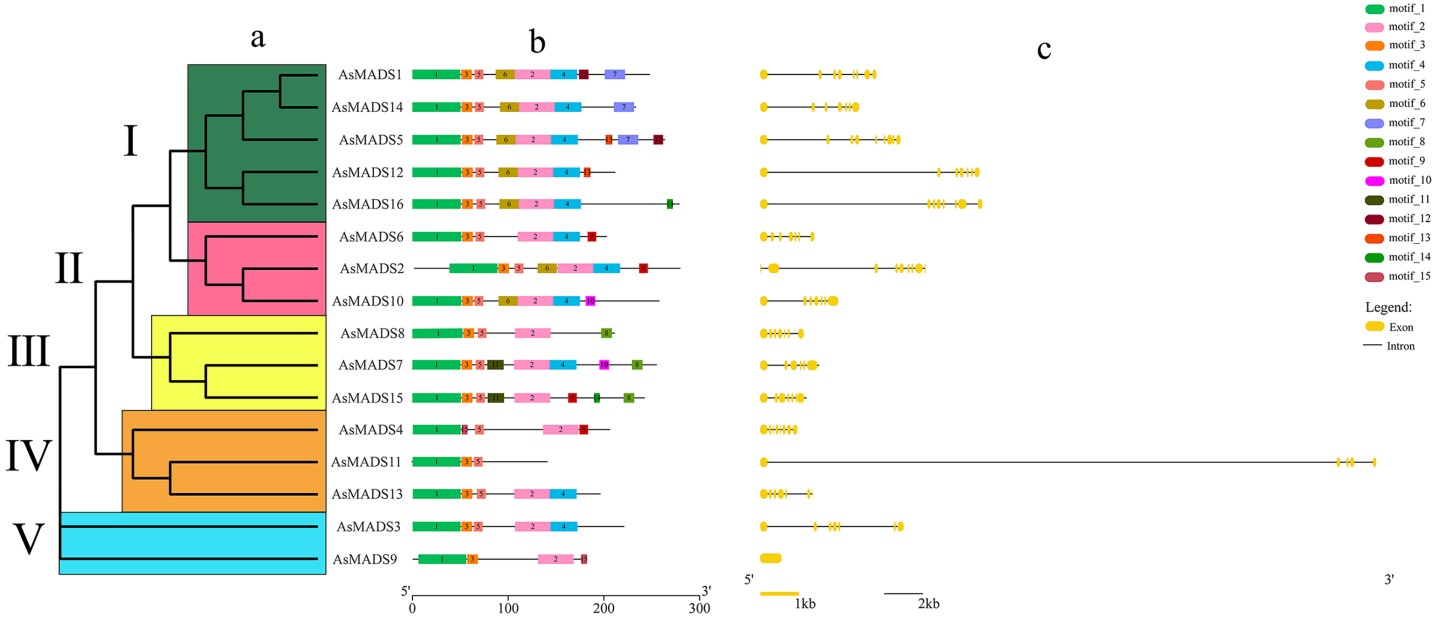

**Figure 3 Gene structure and architecture of conserved protein motifs in AsMADSs.** (A) The phylogenetic tree was constructed based on the full-length sequences of oat MADS family proteins using MEGA-7 software. (B) The motif compositions of AsMADSs. Different colored boxes display different motifs. (C) The exon-intron structure of AsMADSs. Yellow lines indicate CDSs, and black lines indicate introns.

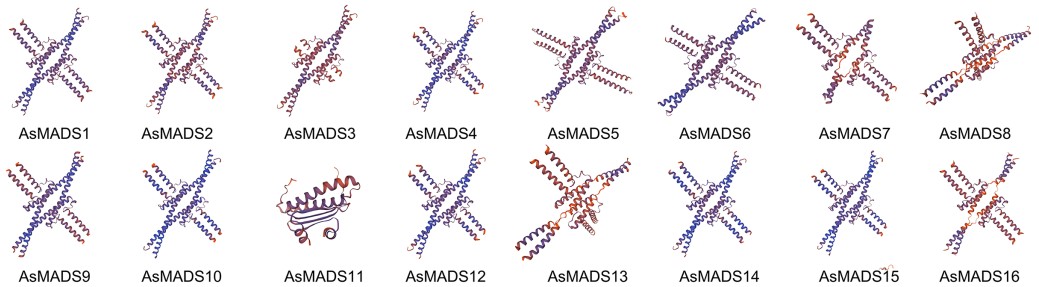

**Figure 4 Predicted three-dimensional domains of 16 AsMADS proteins from oat.**

distributed, indicating that they might be more conserved. Interestingly, motif 11 only existed in the BS subfamily (*AsMADS7* and *AsMADS15*), which may be related to its evolutionary history (Fig. 3B and File S3).

## Chromosome localization and collinearity analysis of *AsMADS* genes

Chromosomal distribution of the *AsMADS* gene family was visualized using TBtools and the oat genome annotation information. The 16 *AsMADS* genes were located on chromosomes 1D, 2D, 3D, 4D, 6D, 7D, 1A, 3A, 6A, 7A, 2C and 6C (Table 1). Briefly, chromosomes 1D, 2C, 3A, and 4D harbored 2 *AsMADS* genes, whereas the other chromosomes possessed only one

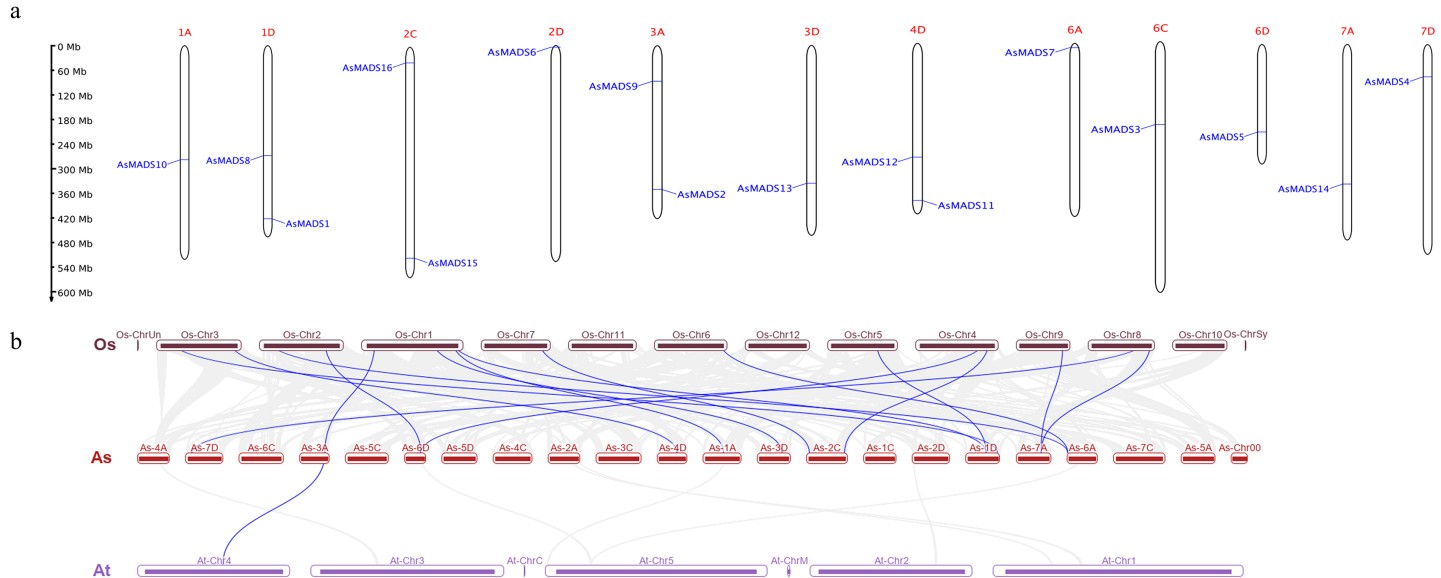

**Figure 5 Chromosomal location and collinearity analysis of 16 AsMADS genes.** (A) Chromosomal location of 16 AsMADS genes in oat. (B) Collinearity analysis of AsMADSs between oat, Arabidopsis, and rice. the Os, As and At respectively indicates that *Oryza sativa*, *Avena sativa*, *Arabidopsis thaliana*.

*AsMADS* gene. Additionally, most members were located on the ends of the chromosome, except for *AsMADS8* and *AsMADS10* (Fig. 5A).

The intraspecific and interspecific collinearity of MADS-box genes was visualized using TBTools. There was no collinearity among the 16 *AsMADS* genes on the oat chromosomes. The interspecific collinearity analysis among Arabidopsis, rice and oat showed that there were 16 pairs of collinear genes between oat and rice, and one gene was syntenic among oat (*AsMADS2*), rice (*Os01g10504.1*) and Arabidopsis (*AT4G18960*), which was annotated as AG (Fig. 5B). The results showed that there was a high degree of conservation and consistency in the linear relationship of MADS-box gene evolution between oat and rice.

## Prediction of *cis*-elements in the promoters of *AsMADS* genes

The 2 kb upstream region of the *AsMADS* genes was extracted by TBtools, and submitted to the online website plantregmap to search for *cis*-elements. The top 12 putative *cis*-elements in the *AsMADS* promoters are shown (Fig. 6). In addition to typical promoter elements, such as TATA boxes and CAAT boxes, G-box, circadian, MYB, MYC, STRE, ABRE, as-1, CGTCA-motif and TGCAG-motif elements were predicted. The promoters of five *AsMADS* members contained circadian elements, namely, *AsMADS1*, *AsMADS7*, *AsMADS8*, *AsMADS11* and *AsMADS15*. Fourteen of the 16 members contained the light-responsive element G-box, except *AsMADS9* and *AsMADS15*, but *AsMADS9* contained other light-responsive elements, such as the I-box and sp1 (File S5). These results indicated that the promoters of all *AsMADS* members contained light-responsive *cis*-acting elements.

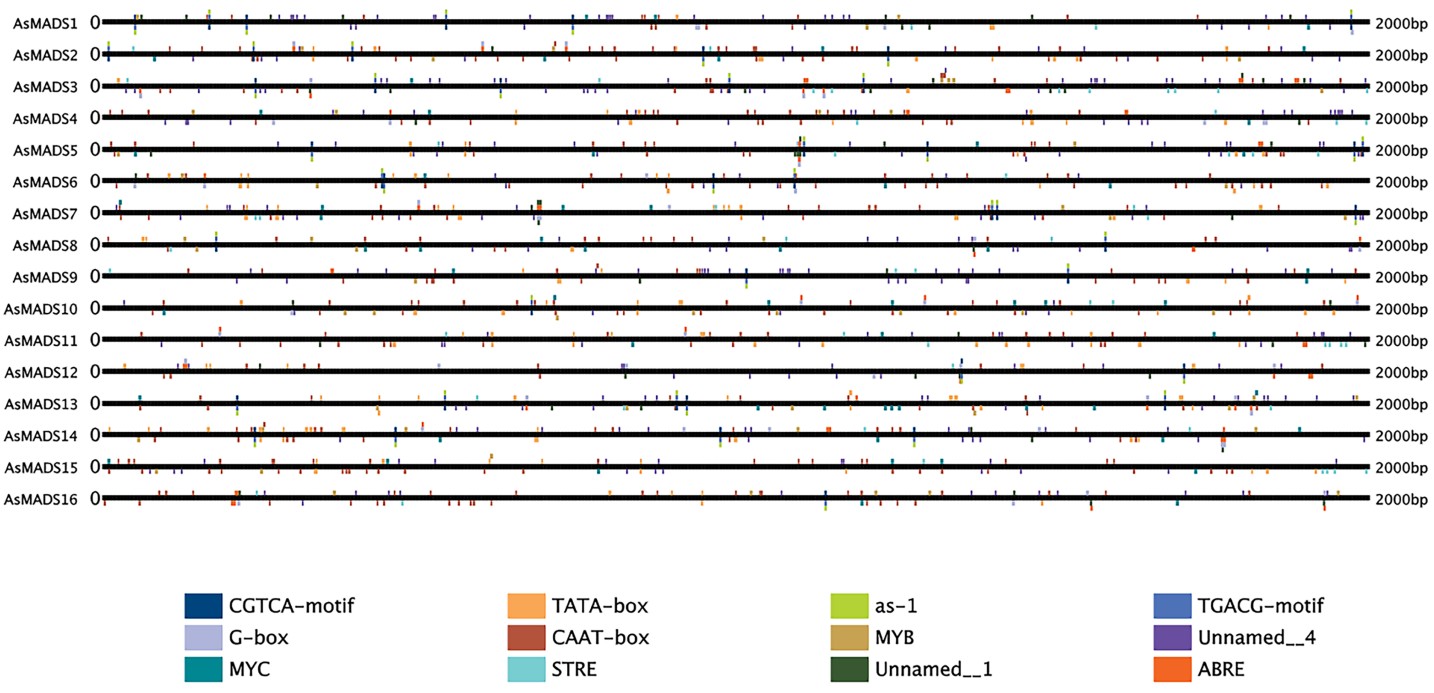

**Figure 6** **Prediction of *cis*-elements in the 2 k upstream regulatory regions of AsMADS genes.** Different colored boxes indicate different *cis*-elements.

## GO enrichment and transcriptome expression analysis of *AsMADS* genes

For the GO classification, the 16 *AsMADS* genes were categorized into three main categories: biological processes, cellular components and molecular functions (Fig. 7). *AsMADS* genes categorized as flower development (GO:0009908), floral meristem determinacy (GO:0010582), floral whorl structural organization (GO:0048459), floral organ formation (GO:0048449), and rhythmic process may be related to the photoperiod insensitivity of oat (File S6).

Based on previous transcriptome data from oat, a heatmap of the *AsMADS* gene expression levels at the initial stage of oat panicle differentiation in HQ2 and MSY4 under short days was drawn. Ten of 16 *AsMADS* members were differentially expressed in the transcriptome data. No matter in panicles or leaves, *AsMADS2*, *AsMADS11*, and *AsMADS16* were upregulated in MSY4, and *AsMADS3* and *AsMADS12* were downregulated in MSY4 (Fig. 8).

## Expression analysis of *AsMADS* genes in HQ2 and MSY4 under short-day conditions

In order to explore the expression specificity, the expression of *AsMADS* gene among materials with different photoperiod sensitivities was carried out. In HQ2, the panicle can only develop to the branch differentiation stage under short days, whereas in MSY4, it can complete the whole panicle differentiation process (*An et al., 2018*). Gene expression data for qPCR are listed in File S8. The expression levels of MSY4 and HQ2 were compared in

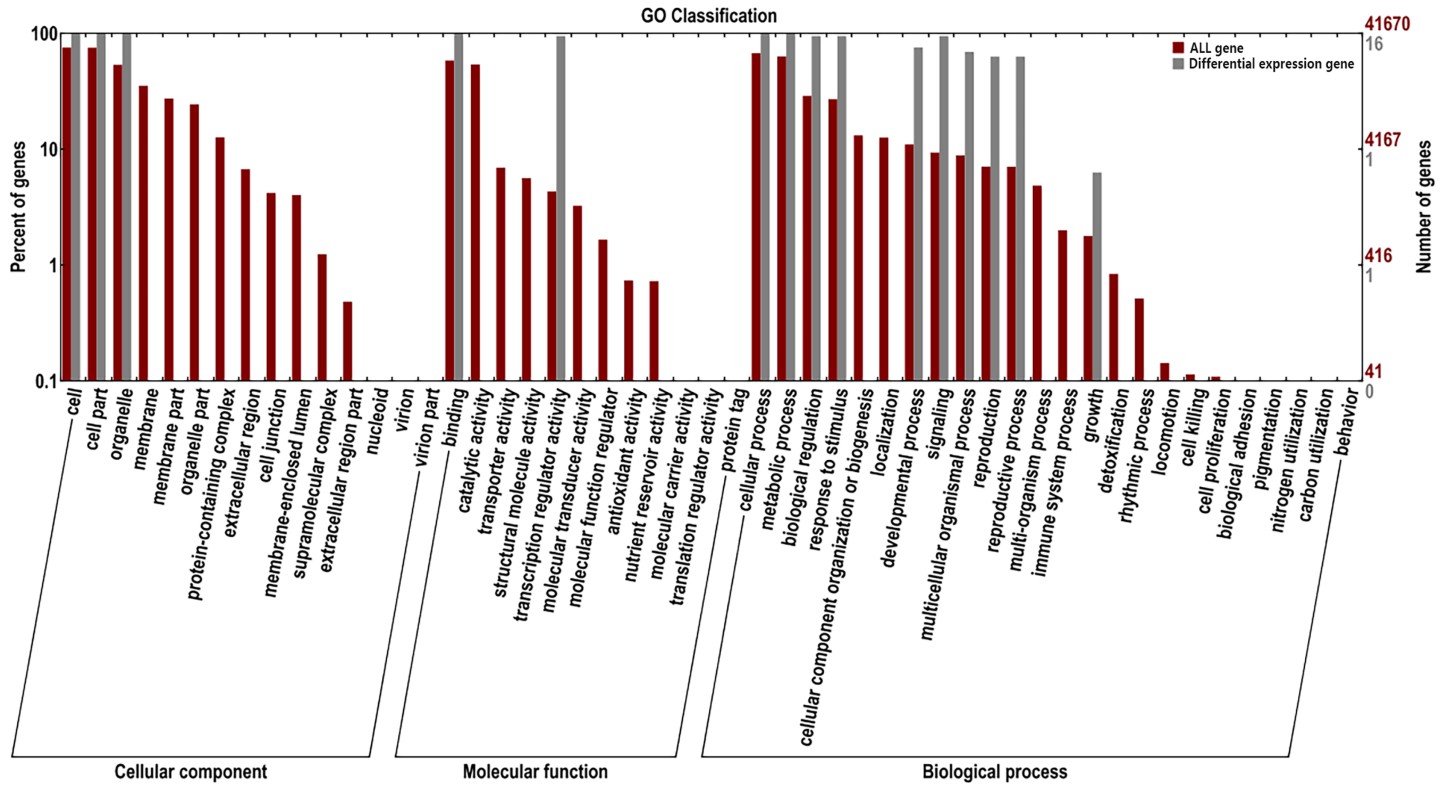

**Figure 7 Gene ontology analysis of 10 AsMADS genes from RNA-seq data.**

the first three panicle differentiation stages. Compared with HQ2, *AsMADS3*, *AsMADS8*, *AsMADS11*, *AsMADS13*, and *AsMADS16* were upregulated at these three differentiation stages in MSY4, while *AsMADS12* was downregulated. *AsMADS6* and *AsMADS9* were downregulated in the initial stage and elongation stage and then upregulated in the branch differentiation stage in MSY4 than in HQ2, while *AsMADS2* was downregulated in the initial stage and showed increased expression in the following two stages. The *AsMADS15* expression level remained basically unchanged throughout the panicle differentiation process between MSY4 and HQ2. With the development of the panicle in HQ2, the expression levels of the *AsMADS3, AsMADS8, AsMADS12* and *AsMADS13* genes gradually decreased (Fig. 9).

In order to study the expression profile of *AsMADS* genes in MSY4 under short-day conditions during the entire panicle differentiation stages, real-time fluorescence quantitative analysis was carried out. With the development of the panicle in MSY4, the expression levels of the *AsMADS9* and *AsMADS11* genes gradually decreased. The expression levels of *AsMADS15* and *AsMADS16* were basically unchanged. The expression levels of the other six members fluctuated at different stages of panicle differentiation (Fig. 10).

## DISCUSSION

Oat is a long-day crop, which limits the expansion of its growing area. Using traditional breeding methods to change the photoperiod sensitivity of oats makes it possible to expand

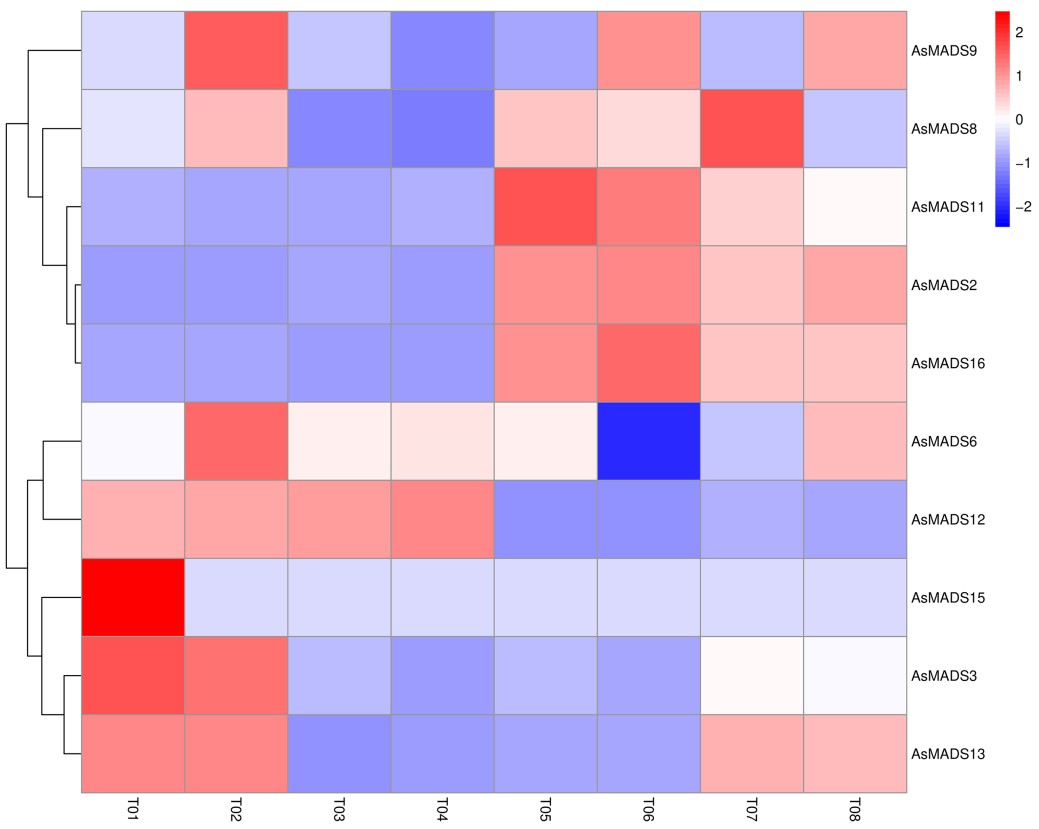

**Figure 8 Expression profiles analysis of 10 AsMADS genes from RNA-seq data.** Differentially expressed genes are screened in panicles (T01, T02, T07, and T08) and leaves (T03, T04, T05, and T06) at the oat initial differentiation stage of HQ2 and MSY4 under short-day conditions. Note: T0-T04 is HQ2. T05-T08 is MSY4. T01, T02, T07, and T08 is panicle. T03, T04, T05, and T06 is leaf.

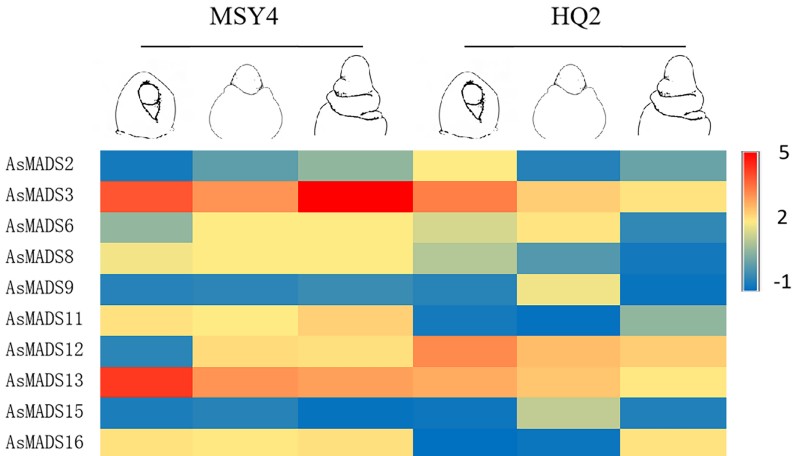

**Figure 9 Gene expression of 10 AsMADS genes in the initial stage, elongation stage and branch differentiation stage of HQ2 and MSY4 under short days from qPCR.** Note: qPCR data results of 10 AsMADS genes was normalized by z-score. Heatmap is performed in Metware cloud platform tool (https://cloud.metware.cn/#/tools/tool-form?toolId=168). Each bar represents a gene. Red indicates that the gene is up-regulated, and blue indicates that the gene is down-regulated.

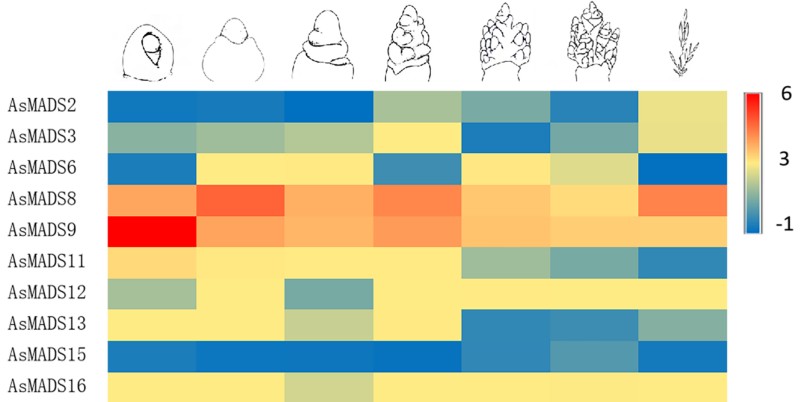

**Figure 10 Expression heat map of 10 AsMADS genes at different developmental stages of MSY4 under short days from qPCR.** Note: qPCR data results of 10 AsMADS genes was normalized by z-score. Heatmap is performed in Metware cloud platform tool (https://cloud.metware.cn/#/tools/tool-form?toolId=168). Each bar represents a gene. Red indicates that the gene is up-regulated, and blue indicates that the gene is down-regulated.

the oat planting area, accelerate the breeding rate, and increase the yield of oats. To date, photoperiod-insensitive oats have been created, but the underlying mechanism is unclear. Based on a previous transcriptome study (*An et al., 2020*), many candidate genes were found, so we focused on *MADS* genes due to their function in flower development and expression differences in transcriptome data.

The release of oat genome data allowed us to identify the MADS family at the genome level. Sixteen members were identified (Table 1) and classified into 9 subfamilies (Fig. 2), including the A-type, C/D-type and E-type genes in the ABCDE model. AP1/SQUA-like (A-type) genes can promote the formation of floral meristems (*Ferrándiz et al., 2000*). *OsMADS14* and *OsMADS18* are AP1/SQUA-like genes in rice. Overexpression of *OsMADS14* can shorten the heading period of rice and affect the formation of floral meristems, and overexpression of *OsMADS18* can cause early flowering (*Jeon et al., 2000*). The E-type usually forms MADS-box protein complexes with A-, B-, C- and D-type proteins to work together to affect floral organ morphology (*Favaro et al., 2003*). A total of 4 E genes (*SEP1/2/3/4*) were identified in Arabidopsis, and 5 SEP genes, namely, *OsMADS1*, *OsMADS5*, *OsMADS24*, *OsMADS34* and *OsMADS45*, were identified in rice (*Pelucchi et al., 2002*). *OsMADS1* is the most detailed SEP-like gene, and it plays a key role in the morphogenesis of floral organs (*Prasad et al., 2001*). The SVP gene controls flowering time in most dicot plants and is an inhibitor of flowering (*Hartmann et al., 2000*). However, the expression levels of *AsMADS8* and *AsMADS9* in the first seven stages of panicle development were also significantly higher than those of other *AsMADS* genes (Fig. 10). All of results indicated that SVP genes promoted the process of panicle development in MSY4 under short-day conditions.

Among relative species of oat, the numbers of *MADS* gene family members in wheat, barley, foxtail millet, maize and brachypodium were 300 (*Raza et al., 2021*), 34 (*Kuijer et al., 2021*), 89 (*Lai et al., 2022*), 211 (*Zhao et al., 2021*) and 57 (*Wei et al., 2014*), respectively. The expansion and contraction of gene families is the result of the interaction

between the plant and the external environment. The reason why the number of MADS genes in oats is less than that in other species may be the result of selection of this gene family during the evolution of oats, or the phenomenon of expansion in other species. The specific reasons need to be further studied.

Plant MADS proteins were conserved (*Favaro et al., 2002*; *Kuijer et al., 2021*; *Zobell, Wolfram & Heinz, 2010*), and MADS proteins of oat were also conserved according to the intron/exon pattern, motif components and predicted 3D structures (Figs. 3 and 4). *AsMADS9* and *AsMADS11* are type I MADS genes, and their gene structures are completely different. *AsMADS9* contains only one exon and no introns, while *AsMADS9* has 5 exons and a very long intron (Table 1 and Fig. 3). Analysis of *cis*-elements revealed that all *AsMADS* genes contained light-responsive elements (Fig. 6). Therefore, the expression of *AsMADS* genes may be activated or inhibited by short-day conditions during panicle development in oats. Non-vernalized spring wheat grown under a short-day photoperiod accumulates *VEGETATIVE TO REPRODUCTIVE TRANSITION 2* (*TaVRT2*) and shows a delay in flowering, suggesting that *TaVRT2* is regulated independently by photoperiod (*Kane et al., 2007*). *GmAGL1* was much more effective at promoting flowering under long-day conditions than under short-day conditions and *GmAGL1* overexpression not only resulted in early maturation but also promoted flowering and affected petal development (*Zeng et al., 2018*). We found that the expression levels of the *AsMADS3*, *AsMADS12*, and *AsMADS13* genes gradually decreased with the development of the panicle in HQ2, and the expression levels of *AsMADS3*, *AsMADS11* and *AsMADS13* were higher in MSY4 than in HQ2 (Fig. 9). The expression levels of *AsMADS6, AsMADS8, AsMADS12*, and *AsMADS13* fluctuated with panicle development (Fig. 10). *AsMADS2, AsMADS8, AsMADS12* and *AsMADS16* belonged to the AG, SVP and SQUA subfamilies (File S3). The main genes that promote MSY4 panicle development under short-day conditions may be SVP, SQUA and Mα genes. In short, high expression of *AsMADS* genes in MSY4 promoted panicle development, while low expression in HQ2 resulted in the arrest of panicle development.

Combined with previous transcriptome studies, the photoperiod pathway of MSY4 in response to short days was predicted. After the seedlings were unearthed, the expression of photoreceptors changed upon sensing inductive sunlight conditions (*Takano et al., 2005*). The expression of *PhyB* increased, and the expression of *PhyC* and *Cry1* decreased. Photoreceptors transmit light signals to circadian genes. The expression levels of *CCA1* and *PRR* genes both decreased. The expression of *GI* decreased, as it is regulated by the circadian clock and can affect the magnitude of circadian rhythm changes. The expression of *CO* in the phloem activates the flower-flowering pathway integration gene *FT* and initiates plant flowering. However, *FT* was not differentially expressed in the initial stage. The factor that actually causes MSY4 to flower under short-day conditions may lie in the process of floral meristem determination. The FT protein is transported from the leaf to the apical meristem cytoplasm to interact with the 14-3-3 protein and then enters the nucleus together with the transcription factor FD protein to form the florigen activation complex and activate the downstream floral meristem genes (*Taoka et al., 2011*). The MADS-box plays an important regulatory role in the developmental process of

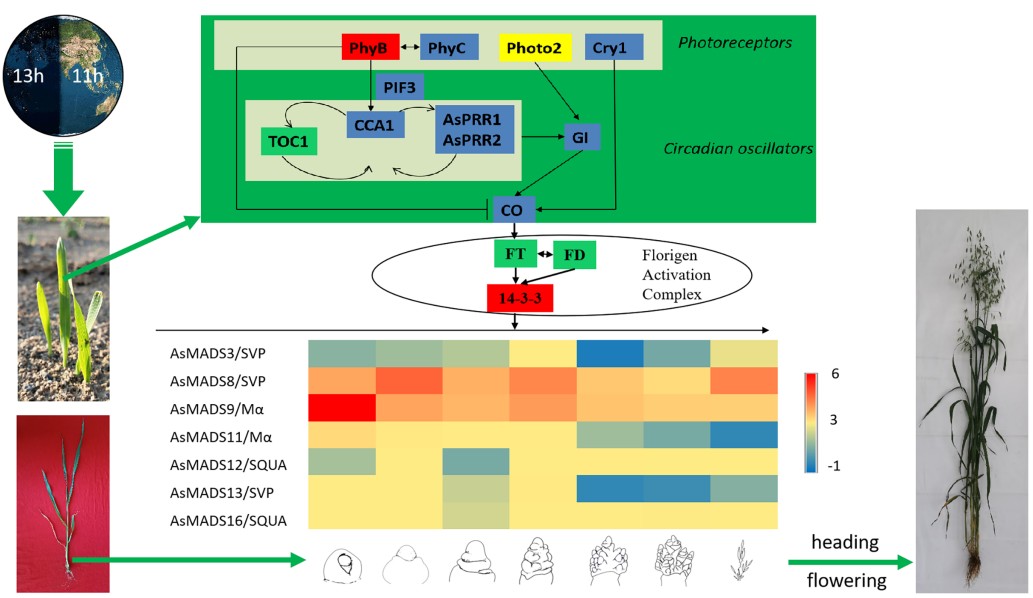

**Figure 11 Predicted photoperiodic pathway of MSY4 under short-day conditions.** Red boxes indicate up-regulation, blue boxes indicate down-regulation, yellow boxes indicate inconsistent expression patterns of multiple transcripts, and green boxes indicate unchanged expression levels.

determining the characteristics of floral meristems. In this study, multiple MADS-box genes were detected to be highly expressed in the panicles of MSY4. In particular, the high expression of the MADS-box genes in the SVP, SQUA and Mα subfamilies regulate the heading and flowering of MSY4 under short-day conditions (Fig. 11).

## CONCLUSIONS

In this study, 16 *AsMADS* genes were identified from the oat genome and could be divided into nine subfamilies. The structures of *AsMADS* members were relatively conserved, but there was no collinearity among them. All members contained light-responsive elements and their expression levels were regulated by light. The expression profiles indicated that *AsMADS* genes belonging to the SVP, SQUA and Mα subfamilies mainly regulate the panicle differentiation process of MSY4 under short-day conditions. Our results can be used for the further functional analysis of these *AsMADS* genes in the photoperiod response under short days.

## ACKNOWLEDGEMENTS

We thank the China National Germplasm Bank for providing the photoperiod-insensitive material MSY4.

### Funding

This research was supported by the "National Key Research and Development Program project (SQ2021YFE010558); Key Laboratory of Wheat Germplasm Innovation and

Utilization Autonomous Region Higher School; Forage Crops and Beneficial Microorganism Germplasm Resources and Molecular Breeding team funding (TD202103)". The funders had no role in study design, data collection and analysis, decision to publish, or preparation of the manuscript.

## Grant Disclosures
The following grant information was disclosed by the authors:
National Key Research and Development Program: SQ2021YFE010558.
Key Laboratory of Wheat Germplasm Innovation and Utilization Autonomous Region Higher School.
Forage Crops and Beneficial Microorganism Germplasm Resources and Molecular Breeding: TD202103.

## Competing Interests
The authors declare that they have no competing interests.

## Author Contributions
- Jinsheng Nan conceived and designed the experiments, performed the experiments, analyzed the data, prepared figures and/or tables, and approved the final draft.
- Jianghong An conceived and designed the experiments, performed the experiments, analyzed the data, prepared figures and/or tables, and approved the final draft.
- Yan Yang conceived and designed the experiments, performed the experiments, analyzed the data, authored or reviewed drafts of the article, and approved the final draft.
- Guofen Zhao analyzed the data, authored or reviewed drafts of the article, and approved the final draft.
- Xiaohong Yang performed the experiments, authored or reviewed drafts of the article, and approved the final draft.
- Huiyan Liu conceived and designed the experiments, analyzed the data, authored or reviewed drafts of the article, and approved the final draft.
- Bing Han conceived and designed the experiments, performed the experiments, analyzed the data, authored or reviewed drafts of the article, and approved the final draft.

## Data Availability
   The raw data are available in the Supplemental File.

## Supplemental Information
Supplemental information for this article can be found online at http://dx.doi.org/10.7717/peerj.16759#supplemental-information.

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
