# Peer review of "Genome-wide identification of the MADS-box gene family in Avena sativa and its role in photoperiod-insensitive oat"

_PeerJ, doi:10.7717/peerj.16759_

## Round 0.1 · original submission · Major Revisions

In light of comments by reviewers, I would suggest the addition of some literature and information (subject to availability) with regard to discussed family in plants other than reference Arabidopsis as well.

Good luck with the revision.

·

Basic reporting

no comment

Experimental design

no comment

Validity of the findings

no comment

Additional comments

Q1- In the result section, Please mention the method briefly in the introductory line of each result.
Q2- Is it possible to annotate the identified motifs? If yes, please mention the already published motifs with their importance in the discussion. 314-315 lines are the results rewritten again, please discuss them by using previous studies.
Q3- As it is described, only one AsMADS gene is collinear with Arabidopsis, have you identified which gene it is? has it showed collinearity with the rice genome as well? Discuss (line 245)
Q4- Please summarize the result section of ABSTRACT. It is very lengthy for example; lines 39-47
Q5- In the Sequence alignment and phylogenetic analysis section of methods: please add oat with other plant species used.
Q6- Please mention Supplementary File 8 in the appropriate section

-Line 85: please check the citation style
-Line 203,205: please check, the protein names should not be italicized.
-Line 70, 78, 82, 91,95, 114, 127, 134, 152, 210, 244, 245, 306, figure 5b legend: Arabidopsis should not be italicized. Please check throughout the manuscript as well
-Please make AsMADS genes writing style consistent throughout the manuscript
-Line 205, 217, 315: AsMADS11 or AsMADS13? Please verify and make it consistent
-Line 221: Figure 3c not 3a
-Line 228: Elaborate the difference
-Line 243: Please mention figure number
- Line 295: Which previous study is referred here and why?
-Line 310-312 line: Are AsMADS8 and AsMADS9 both SVP genes? if not then clarify the line-311 starting with “This result”.
-Line 320-326 lines: The result is promising; can you find similar findings in any other published study if yes please cite to support your data.
-Line 127: Please mention the databases for Arabidopsis, rice and oat
-Line 130 line: Please replace “above results” with an appropriate word.
-Line 133 line: Please replace MADS genes with AsMADS genes
-Line 121 line: Please write “The information of”

In the figures section:
1-Please draw the bootstrap values in Fig 2
2-Please describe the Os, As and At in the 5b figure legend
3-The figure 7 legend text contains “(T01, T02, T07, and T08) and leaves (T03, T04, T05, and T06) at the oat initial differentiation stage of HQ2 and gp012 under short day conditions,” information is not used in the manuscript. Describe and Please mention it in the relative section/s.
4-Please provide high quality of figure 7. The labels are not readable.
5-In Figures 8 and 9, please mention in the legends how you obtained the expression values to generate the heat maps
6-In figure 10, please adjust the size of panicle differentiation stages to the heat map corresponding boxes and label them.
Good luck

·

Basic reporting

1. In this work, the author performed a comprehensive genome-wide identification and characterization of MADS gene family in oat and investigated the expression patterns in various developmental stages. Although the study has been conducted well, I have the following concerns, which authors should address.
2. Some figures are not clear enough, especially the figure 7.
3. There should be a blank space before all the left brackets. And all the genes should be italic.
4. Some paragraph is too short, like L136-L138, authors can merge it with next paragraph.

Experimental design

5. The number of MADS genes is much less than Arabidopsis and rice, what’s the number in oat’s close relative? I am curious why there are so few MADS genes in oat. Authors should verify it and discuss the reason in the discussion.
6. L130, L139, in these two paragraphs, the authors used pfamscan and NCBI-CDD to find conserved domains, respectively, I suggest the authors use these two methods in L130 to double check the domains.
7. L157, “in which 10 genes” should be “in which 10 MADS genes”, and remove the next sentence, it’s redundant.
8. MSY4 is the same with gp012? It’s confusing, please state their relationship.
9. The description and display of transcriptome expression analysis are disordered, for example:
a. For figure 7-9, which is from RNA-seq data, which is from qPCR.
b. The legend of Figure 7 is “Differentially expressed genes are screened in panicles (T01, T02, T07, and T08) and leaves (T03, T04, T05, and T06) at the oat initial differentiation stage of HQ2 and gp012 under short-day conditions.”, the color bar means normalized FPKM or log2FC between HQ2 and gp012? That means the figure is from one material or two?? Or it means the differentially expression was detected between panicles and leaves?? But the leaves were not mentioned in the results. And if the GO classification is based on all 16 AsMADS genes, the authors shouldn’t put it after the transcriptome expression analysis.
c. L163 “under 12h days”, is this short day?
d. L276-280, these sentences were also compared with HQ2, or only the expression level in MSY4? If compared with HQ2, the description was not accurate.
e. L281-282, this sentence should be put after the previous paragraph, for they are all about Figure 8.
f. L285, the expression levels of MADS2 and MADS3 were unchanged?
In short, I commend the authors reorganize this part.

Validity of the findings

10. Conclusions are well stated, I think this work will facilitate further functional analysis of MADS genes in oat and provide clues for the breeding.

---

## Round 0.2 · Minor Revisions

Dear Authors, kindly revise as two reviewers have suggested.

Thank you and good luck

·

Basic reporting

No comment

Experimental design

No comment

Validity of the findings

No comment

Additional comments

-Line 35, 99: make the nomenclature HongQi2hao consistent.
153-158: add leaves data too
157: heatmap by which software?

-Please make writing style consistent throughout manuscript. For example, Line 203,210: Figure or Fig?
Line 225: end with predicted is unnecessary.
Line 227: after AsMADS11, there should be comma
Line 248: gene names should be italic
-In the methodology section: the authors have not added the gene ontology method
-Wrong citation style is used, for example line 328, O et al., 2010, it should be (ZOBELL et al., 2010), instead of HNJ et al., 2021, it should be Kuijer et al., 2021. Re-cite every reference in the manuscript. -Similarly, the references need extensive revision for example the authors name, journal name formatting is particularly incorrect/inconsistent.
-Line 304: change “will allow “ to “allowed”
-Based on fig 9, the qpcr data shows , MAD9 is downregulated while fig 10 shows the expression of MAD9 is upregulated please remove or explain the discrepancy in discussion line 317-320.
- FLC (FLOWERING LOCUS C) is a gene that plays a crucial role in regulating the flowering time in plants. In Arabidopsis thaliana, FLC is known to be regulated by vernalization, a process where the plant is exposed to a prolonged period of cold temperature to induce flowering. Some monocots, cereals like wheat has FLC homolog, is there possibility that vernalization in oats is carried out other than FLC pathway? Can you find literature to support your result? And add in discussion part.
Line 330 : type II??
336-338: italic the gene names
341-349: refer the figure
-describe the expression trend differences or similarities between duplicated pairs in results and discussion to draw a conclusion.
As mentioned by authors, The gene was syntenic among oat (AsMADS2), rice (Os01g10504.1) and Arabidopsis (AT4G18960), which was annotated as AG (AGAMOUS, K-box region and MADS-box transcription factor family protein). If the importance of this gene is published realted to same trait?

,

·

Basic reporting

The authors had revised well according to my suggestions.
ps. There are a few mistakes in Figure 7, the title should be 16 AsMADS, not 10? And delete "from RNA-seq data", the right axis DIFF gene means what, the values are 1,1,16?
And the Supplementary_File_6 only supplied Biological Process GO terms, please add CC and MF.

Experimental design

No comment.

Validity of the findings

No comment.

Additional comments

No comment.

---

## Round 0.3 · accepted · Accept

Please edit the legend as suggested by the reviewer.
Congratulations on your effort.
Keep it up!

·

Basic reporting

Generally, differentially expressed genes was marked as DEGs, as authors didn't mention DIFF in the main text, it's better to add a legend.

Experimental design

no comment

Validity of the findings

no comment